# Time-Series Image-Based Automated Monitoring Framework for Visible Facilities: Focusing on Installation and Retention Period

**DOI:** 10.3390/s25020574

**Published:** 2025-01-20

**Authors:** Seonjun Yoon, Hyunsoo Kim

**Affiliations:** Department of Architectural Engineering, Dankook University, 152 Jukjeon-ro, Yongin-si 16890, Republic of Korea

**Keywords:** jack support, monitoring, document information, object detection, optical character recognition (OCR), natural language processing (NLP)

## Abstract

In the construction industry, ensuring the proper installation, retention, and dismantling of temporary structures, such as jack supports, is critical to maintaining safety and project timelines. However, inconsistencies between on-site data and construction documentation remain a significant challenge. To address this, this study proposes an integrated monitoring framework that combines computer vision-based object detection and document recognition techniques. The system utilizes YOLOv5 for detecting jack supports in both construction drawings and on-site images captured through wearable cameras, while optical character recognition (OCR) and natural language processing (NLP) extract installation and dismantling timelines from work orders. The proposed framework enables continuous monitoring and ensures compliance with retention periods by aligning on-site data with documented requirements. The analysis includes 23 jack supports monitored daily over 28 days under varying environmental conditions, including lighting changes and structural configurations. The results demonstrate that the system achieves an average detection accuracy of 94.1%, effectively identifying discrepancies and reducing misclassifications caused by structural similarities and environmental variations. To further enhance detection reliability, methods such as color differentiation, construction plan overlays, and vertical segmentation were implemented, significantly improving performance. This study validates the effectiveness of integrating visual and textual data sources in dynamic construction environments. The study supports the development of automated monitoring systems by improving accuracy and safety measures while reducing manual intervention, offering practical insights for future construction site management.

## 1. Introduction

The construction industry operates within complex workflows and stringent safety standards [1]. As a result, continuous monitoring is essential to ensure project success and to identify potential issues early [2,3,4,5,6,7,8]. While effective construction process monitoring is critical, conventional manual monitoring methods face limitations in meeting the demands of modern construction sites due to the complexity of projects and limited on-site personnel [3,4,9,10,11,12]. As projects grow in scale and complexity, the limitations of these manual approaches become increasingly apparent [13,14,15,16].

In this context, there is a growing demand for automated monitoring solutions capable of managing complex sites more efficiently [2,17,18]. With advances in computer vision technology, construction sites are transitioning to automated systems that monitor site conditions using visual data [19,20,21,22,23,24,25]. Image-based data, collected from various devices such as fixed cameras, is processed using advanced technologies, enabling real-time analysis [4,26,27]. This allows for the early detection of potential risks and helps ensure compliance with worker safety regulations rather than merely supporting predefined safety standards [20]. These automated monitoring systems provide continuous oversight of work progress, enabling managers to assess site conditions quickly [7,28,29]. Additionally, they allow for a more proactive approach to managing project workflows compared to traditional manual methods [28,30].

Despite these technological advancements, existing image-based monitoring systems face limitations in fully capturing the dynamic nature of construction sites [24,31,32]. Temporary structures such as jack supports are frequently relocated and adjusted as project phases progress, making it challenging to track them by analyzing individual frames alone [11,33]. Among the various vision-based monitoring approaches, methods that rely on capturing fixed points in time have shown effectiveness in detecting specific changes but often fail to comprehensively reflect the dynamic and complex workflows of construction sites [24,34]. Recent studies have demonstrated that workflow-based continuous monitoring techniques are more effective in capturing the evolving characteristics of construction processes and provide a more comprehensive perspective on site activities [35,36,37]. However, existing automated monitoring approaches that use computer vision often lack integration between on-site image data and essential documents, such as drawings and work schedules [25,38,39]. This lack of coordination can lead to inconsistencies between site data and documentation [25,40], potentially resulting in structural defects or safety issues when unexpected changes occur [41]. For example, if a detected structure on-site does not match the documented installation requirements, this can lead to inconsistencies between the actual site data and the information recorded in the documentation. Such inconsistencies may result in errors or misunderstandings about the structural setup [37]. Thus, a system that can compare on-site data with construction drawings and project plans is necessary [35,42].

To address these challenges, this study proposes a monitoring framework focused on bridging the gap between on-site data and documented information. This framework combines computer vision technology with document analysis methods, integrating object detection, optical character recognition (OCR), and natural language processing (NLP) to monitor temporary structures in complex environments. Specifically, it aims to ensure that temporary structures, like jack supports, in constrained and structurally complex environments such as underground parking lots, are installed, maintained, and dismantled according to documented design standards. Through this monitoring framework, the overall goal is to improve consistency between on-site conditions and documentation, enhancing the accuracy and efficiency of monitoring on construction sites.

## 2. Literature Review

Efforts to improve monitoring systems for construction sites have seen significant advancements in recent years [8,43]. Current studies have focused on developing and applying various techniques to enhance the accuracy, efficiency, and practicality of monitoring dynamic construction environments [30,43,44]. While these studies provide invaluable insights and tools for improving site management, there are still challenges to address, particularly the inconsistencies between on-site data and documentation [45,46]. This section reviews the state of the art in monitoring systems, their limitations, and the research gaps that this study aims to address.

### 2.1. Advances in Construction Document Monitoring

Recent developments in document recognition technology have significantly enhanced the ability to extract and analyze critical information from construction schedules, drawings, and work plans [47,48]. Optical character recognition (OCR) and natural language processing (NLP) technologies enable the digitization and interpretation of text-based data, facilitating access to essential information such as installation schedules and procedural guidelines [49,50]. For example, OCR tools like ABBYY FineReader 15 software can convert construction documents into structured data, which can then be cross-referenced with real-time field data to verify project requirements and track progress effectively [51,52].

The integration of OCR and NLP with monitoring systems has proven effective in synchronizing documentation and site conditions [53,54,55], enabling dynamic, context-aware monitoring [56,57,58]. For instance, image data of jack supports can be compared with documented placement plans to ensure installations align with design specifications and safety standards [54,59]. However, technological challenges persist, such as variations in data formats, inconsistent document interpretations, and limited flexibility across project phases [32,60]. Addressing these challenges is critical to creating adaptable and robust monitoring frameworks that reflect ongoing construction workflows [61,62,63].

### 2.2. Monitoring Challenges and Vision-Based Solutions

Monitoring dynamic elements on construction sites is inherently challenging [64] due to the frequent relocation and changing conditions of temporary structures [65]. These challenges are compounded by inconsistencies between on-site data and documentation, such as drawings and schedules [66,67,68]. For instance, while on-site data provide real-time insights into the condition of temporary structures [66,69], they often fail to align with the static layouts in design drawings or the predetermined timelines in work schedules [70,71]. These inconsistencies require significant effort to resolve, increasing the risk of inefficiencies, errors, and safety hazards [72].

To address these issues, vision-based monitoring systems have emerged as a promising solution [73,74]. Object detection algorithms such as YOLO and Faster R-CNN, powered by convolutional neural networks (CNNs), have been widely adopted for monitoring temporary structures and equipment [31,36,75,76]. These technologies are particularly effective in identifying hazards and providing warnings to managers [75,77]. However, current vision-based models are primarily designed for static objects, making them less effective for temporary structures that frequently change positions and conditions [78,79]. Continuous monitoring systems, capable of tracking evolving workflows, offer a more comprehensive approach but require further integration with document-based data to ensure consistency with design specifications [72,80].

Integrating vision-based monitoring with construction document recognition presents an opportunity to overcome these limitations [32,81]. By aligning on-site data with documentation, such systems can identify discrepancies [82], improve workflow tracking, and enhance the reliability of site monitoring systems [74,83,84]. This integrated approach represents a significant step toward addressing the dynamic and complex nature of construction environments [85,86].

### 2.3. Integration of Monitoring Systems

To address the limitations identified in previous sections, this study proposes a unified framework that integrates computer vision technology with document recognition methods. This framework aims to align on-site data with documented requirements, addressing inconsistencies caused by the dynamic nature of construction sites. The proposed system combines continuous monitoring with document-based verification. This approach ensures comprehensive monitoring of temporary structures, such as jack supports, across their installation, maintenance, and dismantling phases.

This integration enables a consistent representation of temporary structures by combining on-site data with construction documentation. It ensures that the positioning and condition of temporary structures are accurately reflected in relation to documented specifications. Unlike conventional approaches that focus on static datasets or pre-determined schedules, this framework considers the evolving conditions of construction sites to provide a more holistic perspective on site activities.

While achieving full integration between visual data and construction documents presents challenges, such as variations in data formats and interpretations, this approach offers a systematic foundation for addressing these issues. By bridging the gap between on-site conditions and documentation, the framework contributes to more reliable and efficient monitoring systems tailored to the complexities of modern construction sites.

## 3. Methodology

### 3.1. Research Framework

This section outlines the suggested framework integrating (1) object detection algorithms, (2) construction drawing and document recognition, and (3) jack support monitoring techniques. The primary objective of this framework is to facilitate the automated monitoring of jack supports in construction sites, ensuring their correct installation, retention, and removal in compliance with safety standards and construction plans.

Figure 1 shows the interaction between these components. The object detection algorithm is both used to detect jack support symbols on drawings and on-site jack supports. The model is trained with a dataset of labels consisting of jack support images to accurately detect and identify jack support locations in drawings and on-site images. Drawings and on-site images are preprocessed to enhance clarity and optimize them for analysis, ensuring the accurate detection of the jack support. On-site image data are collected by a camera worn on the chest of construction managers, which continuously monitors the status of the jack support. Alongside this, optical character recognition (OCR) is used to extract document information such as installation and removal times, and natural language processing (NLP) techniques are used to interpret relevant work instructions. Based on this information, the detected data are compared to drawings and documentation to verify where and when the jack support was installed and whether it complies with the retention period. Time-based detections at T_n_ and T_n−1_ monitor the maintenance period and assess whether the jack support complies with the safety standards and recorded requirements.

### 3.2. Data Collection and Preprocessing

In this study, on-site images were captured using a camera mounted on the chest. As the manager moved throughout the site, multiple images of jack supports were collected from various angles to ensure a comprehensive dataset. In addition to the images, construction drawings and work plans were gathered to provide contextual information about the site. The collected data are shown in Figure 2.

The collected data consisted of on-site images, construction drawings, and related documents. The on-site images captured videos of jack supports, while the construction drawings and documents provided detailed plans and schedules for the site. To improve analysis quality, both on-site images and construction drawings underwent several preprocessing steps. These included noise reduction (Figure 2b), contrast enhancement (Figure 2c), alignment correction (Figure 2d), normalization (Figure 2e), and segmentation (Figure 2f), focusing on relevant features like jack supports. For instance, contrast enhancement and noise reduction improved the clarity of construction drawings, ensuring the accurate identification of jack support symbols. Similarly, for on-site images, alignment correction was applied to account for slight shifts in camera positioning as the manager moved, allowing for the consistent detection of jack supports across various angles and lighting conditions. Object detection was utilized twice within this framework, first to detect jack support symbols in construction drawings and second to detect the actual jack supports on-site. YOLOv5 was employed for both tasks, allowing for the precise identification and localization of jack supports in both images and drawings.

A notable distinction of this study lies in its dual application of object detection—not only for on-site images but also for construction drawings—ensuring alignment between planned and executed installations. Unlike traditional studies that apply the same preprocessing methods to all images, this research used specific preprocessing techniques tailored to each image type. This approach improved the accuracy and flexibility of object detection. This approach facilitated better alignment between the construction plans and on-site implementation. Figure 2 also includes a sequence showcasing the original and preprocessed images to demonstrate the effectiveness of applied techniques. This sequence covers blurring, contrast enhancement, alignment correction, normalization, and segmentation steps. By comparing the original images with those that underwent preprocessing, the effectiveness of the techniques can be observed. This combined approach of preprocessing and object detection was consistently applied to both the on-site jack supports and the identification of jack support symbols in construction drawings, ensuring a consistent and comprehensive monitoring process.

### 3.3. Object Detection Algorithm

This section describes the object detection process used in this study, which involves detecting jack support symbols in construction drawings and identifying actual jack supports on-site. The process ensures that jack supports are accurately monitored and verified across both planned drawings and on-site installations.

The first stage focuses on identifying jack support symbols within the construction drawings. To achieve this, the drawings underwent preprocessing steps, such as contrast enhancement and alignment correction, to improve clarity and accuracy. The YOLOv5 model, initially trained on labeled jack support symbols, was applied to these preprocessed drawings. However, a significant challenge arose due to the visual similarity between jack support symbols and other structural elements, such as pillars. To resolve this issue, an additional pillar class was introduced into the dataset, enabling the model to better differentiate jack support symbols from visually similar elements. The dataset for this stage consisted of 800 images, including 500 labeled for jack supports and 300 for pillars. The model was further refined by utilizing distinct visual markers found in the drawings, such as jack support symbols highlighted in red and visually similar pillars marked in orange, as shown in Figure 3a–c. This refinement significantly improved detection accuracy by reducing false positives and ensuring the reliable identification of jack supports within the construction drawings.

The second stage applies the detection algorithm to on-site images captured at the construction site. Initially, the model was trained solely on jack supports, but it misclassified columns as jack supports due to their similar structure. To address this, columns were added as a separate class to the dataset. Furthermore, to account for the dynamic nature of construction sites, a “person” class was also added to the dataset to prevent the misclassification of workers as jack supports. Images were collected using a chest-mounted camera worn by the site manager and were preprocessed using noise reduction and alignment correction techniques. For this stage, the YOLOv5 model was trained on a dataset consisting of 1200 images, with jack supports (500 images), pillars (400 images), and people (300 images). This expanded classification system allowed the model to differentiate jack supports not only from pillars but also from construction workers present in the scene. Figure 3d–f illustrates this process; (d) shows an example of an on-site image captured by the camera, (e) highlights detected jack supports, and (f) demonstrates the successful classification of pillars, ensuring reliable differentiation.

This dual application of object detection ensures that both the planned placements in the drawings and the actual installations on-site are consistently monitored and verified, allowing for any discrepancies to be addressed promptly.

### 3.4. Recognition of Document Information

In this section, the process of extracting critical information from construction work orders is detailed using a combination of OCR and NLP techniques. The primary goal is to identify and compile essential details such as the underground floor level, jack support number, installation date, and dismantling date into a structured list. As illustrated in Figure 4, The first step involves image preprocessing, which enhances the quality of the document images to improve the accuracy of text recognition. These preprocessing techniques, identical to those discussed in the previous section (as shown in Figure 2), include noise reduction, contrast enhancement, and alignment correction, ensuring that the images are in optimal condition for subsequent OCR processing.

Once the images have been preprocessed, OCR is applied to convert the visual text in the document images into machine-readable characters. The extracted text then undergoes several text preprocessing stages to prepare it for further analysis. These stages include Table 1 following:

After text preprocessing, NLP techniques are applied to extract and analyze relevant information in Table 2. These techniques include the following:

Finally, the processed information is output as structured data organized into a clear and usable format such as structured tables or categorized lists. This structured output is essential for systematically tracking and verifying the installation and dismantling of jack supports.

### 3.5. Matching Process Between Drawings and On-Site Detection

To ensure that jack supports detected in construction drawings correspond accurately to those identified on-site, a straightforward matching process was implemented using spatial positions, basic identifiers, and additional floor-level markers. First, the spatial coordinates of jack supports from construction drawings were extracted during the preprocessing stage. These coordinates, represented as simple (x, y) positions, were compared with the approximate positions of jack supports identified in on-site images. Instead of relying on complex calibration methods, a direct comparison was performed based on proximity, where detections within a certain distance (e.g., 1 m) were considered a match.

In addition to spatial matching, the process incorporated basic floor-level labels and area markers to address ambiguities in similar configurations. For example, labels such as “Floor 2, Section A” or simple numbering systems were added to the drawings and referred to during on-site inspections. These markers provided an additional layer of reference to distinguish jack supports in complex layouts without requiring advanced algorithms.

To handle potential mismatches or missing data, a manual verification process was introduced. When the system failed to detect a match or flagged potential errors (e.g., misclassification of a column as a jack support), the data were reviewed by referring to construction drawings and on-site photographs. This simple yet effective method ensured that the detected jack supports were consistently and accurately linked between drawings and the site.

### 3.6. Verification of Jack Support Installation

In this section, the process of determining whether a jack support has been correctly installed is explained. This method involves comparing on-site detection data with the planned installation locations from construction drawings and the documented installation times. For example, at a specific time T_n_, the system might detect jack support_1 while failing to detect jack support_2, even though both jack supports are indicated on the construction drawing. By referencing the installation timelines extracted from the documentation, the system can assess whether jack support_2 should have been installed by that time.

The provided Figure 5 visually illustrates this process. The first part of Figure 5 shows the planned installation locations of jack support_1 and jack support_2 based on construction drawings. The second part depicts the on-site detection process, where jack support_1 is detected but jack support_2 is not. The third part represents the installation timeline extracted from the documents, which is then used to determine whether the undetected jack support should have been installed at the time of measurement. This integrated approach ensures that discrepancies between planned and actual installations are promptly identified and addressed.

### 3.7. Verification of Jack Support Retention Compliance

In this section, this study details the process used to determine whether the jack supports on-site have complied with their designated retention periods, as specified in the construction documents. This is achieved by comparing the on-site detection data of the jack supports with the documented retention periods extracted from construction documents.

For example, during monitoring at various time intervals, the system might first detect jack support_3 at time T_1_. This detection continues through subsequent intervals, T_2_ to T_6_, indicating that the jack support_3 remains in place. However, at time T_7_, jack support_3 is no longer detected, marking the end of its on-site retention period, denoted as T_7_–T_1_. This on-site retention period is then compared with the retention period documented in the construction records. If the on-site retention period (T_7_–T_1_) is greater than or equal to the documented period, jack support_3 is considered to have complied with the required retention period. If it is shorter, this indicates non-compliance. This process is visually depicted in Figure 6. The first part of Figure 6 shows the timeline of observations, starting from T_1_ when jack support_3 is first detected and continuing through T_2_ to T_6_, where jack support_3 remains detectable. By T_7_, jack support_3 is no longer detected. Figure 6 also highlights the comparison between the documented retention period and the on-site retention period to assess compliance.

## 4. Results

### 4.1. Object Detection Results

This section presents the results of the object detection algorithm applied to both the construction drawings and on-site images. The results include an analysis of performance metrics such as the F1 confidence curve, precision–recall curve, recall–confidence curve, and precision–confidence curve, along with the evaluation of accuracy through the confusion matrix and object detection outputs.

#### 4.1.1. Jack Support Detection on Drawing

The object detection algorithm was first applied to detect jack support symbols on the construction drawings. The YOLOv5 model was used for this task after preprocessing the drawings to enhance clarity and accuracy. The model’s performance was evaluated using various metrics, as illustrated in Figure 7. The F1 confidence score reached approximately 0.97, and the precision–recall, recall–confidence, and precision–confidence scores consistently maintained values of 0.99 across varying confidence levels.

The accuracy of the model in detecting jack support symbols was further evaluated using the confusion matrix, as shown in Figure 7. The confusion matrix revealed a true positive rate (TP) of 0.97, true negative rate (TN) of 0.99, and false positive (FP) and false negative (FN) rates of 0.01 and 0.03, respectively.

Based on these values, the accuracy of the detection was determined using the following formula:$$\frac{TP+TN}{TP+TN+FP+FN}$$

Using the above formula, the model’s accuracy was calculated to be approximately 98%.

#### 4.1.2. On-Site Jack Support Detection

The object detection algorithm was next applied to on-site images captured at the construction site to detect the installed jack supports. Similarly to the process described in Section 4.1.1, the YOLOv5 model was employed for this task after performing appropriate preprocessing on the images to enhance clarity and detection accuracy. The performance of the model was evaluated using the same four metrics, namely the F1 confidence curve, precision–recall curve, recall–confidence curve, and precision–confidence curve, as shown in Figure 7. These curves help assess the model’s accuracy and robustness across different confidence thresholds. Additionally, a confusion matrix is provided in Figure 7 to illustrate the breakdown of predictions.

The confusion matrix for this section divides the results into four categories, which are true positives (TPs), true negatives (TNs), false positives (FPs), and false negatives (FNs). In this context, TP refers to correctly identified jack supports, TN refers to correctly identified non-jack support elements (like pillars and the background), FP refers to elements incorrectly detected as jack supports, and FN refers to jack supports that were not detected by the model. These terms have the same definitions as in Section 4.1.1. The accuracy was calculated similarly to Section 4.1.1 based on the values in the confusion matrix. The formula used to compute the accuracy remains the same, and it was found that the model achieved approximately 95.8% accuracy in detecting jack supports on-site.

Additionally, Figure 8 presents the detection results captured at the construction site. In these on-site images, the model successfully detected jack supports, pillars, and persons. Figure 8 shows the use of color coding to differentiate between the detected classes, with red for jack supports, orange for pillars, and pink for persons.

### 4.2. Document Recognition Result

The document recognition process focused on extracting and organizing essential information from the construction work orders (checklists) into structured lists, particularly related to jack support installation and dismantling times. A total of 17 checklists were processed, and the object detection and OCR algorithms successfully recognized the necessary information from all checklists. The consistent format of the checklists, with fixed locations for the jack support number, installation date, and dismantling date, contributed to the high accuracy of the recognition process.

As seen in Figure 9, the critical sections of the checklist include the checklist number, underground level, jack support number, installation time, and dismantling time. The red highlighted boxes show the locations where this information was extracted. The accuracy of the extraction process was verified by comparing it to manually labeled data.

The results of this recognition process are summarized in Table 3, which presents a table indicating the jack support numbers recognized from each checklist, along with the corresponding installation and dismantling times. Table 3 shows that all the required information was successfully extracted.

### 4.3. Jack Support Installation Determination

This section evaluates the installation status of the 23 jack supports, where the installation time for all units was standardized to T_1_ (12 February 2024), which marks the first inspection date. The objective was to assess whether the jack supports were installed in accordance with the project plan by integrating data from three primary sources, namely (1) the detected jack support symbols from the construction drawings, (2) the detection results captured on-site, and (3) the installation timelines extracted from work order documents. Table 4 summarizes the detection results of the 23 jack supports. It presents whether each jack support was successfully detected both in the construction drawings and through on-site detection at T_1_. As shown in Table 4, 22 jack supports were correctly identified in both sources, while JS_10 failed to be detected during the on-site inspection.

During the initial inspection on 12 February 2024, the on-site object detection system successfully detected 22 out of 23 jack supports, yielding a detection rate of 95.6%. However, the system encountered misclassification issues. The issue involved JS_10, where a pipe was erroneously classified as a jack support, as shown in Figure 10.

### 4.4. Determining Retention Compliance

This section evaluates the compliance of the 23 jack supports with their designated retention periods. All jack supports were installed on 12 February 2024 (T_1_), which also marks the first inspection date. Inspections were conducted daily for 28 days to monitor the presence of the jack supports. The default dismantling time was set at 28 days after installation. However, some jack supports were scheduled for dismantling 7 or 14 days post-installation, simulating varied construction timelines.

Table 5 outlines the detection status of each jack support from T_1_ to T_28_. For each jack support, Table 5 records the planned dismantling time and daily detection status. ○ indicates successful detection on-site, while × signifies instances where the jack support was not detected. The objective of the inspection was to ensure that each jack support remained in place until its scheduled dismantling date and was not detected afterward.

Three notable errors were observed during the retention compliance period as follows:(a)JS_10 detection error: As previously mentioned in Section 4.3, the system falsely identified a ceiling pipe as JS_10. This misidentification persisted from T1 through T28, leading to a complete failure in detecting JS_10 throughout the retention period. This error is highlighted in Table 4 and Table 5 and corresponds to Figure 11a.(b)JS_14 detection error: JS_14 was correctly detected until the dismantling of JS_16, which occurred on T14. From T15 onwards, the system erroneously identified a wall corner as JS_14, leading to missed detections until T28. This error was influenced by the removal of JS_16, which altered the spatial configuration of the surrounding objects. Figure 11b illustrates the misclassification, where the corner of a pillar was incorrectly labeled as JS_14.(c)JS_20 detection inconsistency: JS_20 experienced inconsistent detection due to varying lighting conditions on-site. The system sometimes detected JS_20 correctly, while at other times, it failed to do so. These fluctuations in detection were primarily observed until its dismantling on 26 February 2024 (T14). After T14, when JS_20 was dismantled, it was no longer detected as expected. Figure 11c provides a comparison of detection differences caused by lighting changes.

In summary, while most jack supports adhered to the expected retention periods, the system encountered detection issues in three instances, namely JS_10 (due to the misidentification of a pipe), JS_14 (misclassified as a wall corner after the removal of JS_16), and JS_20 (inconsistent detection influenced by lighting conditions).

To address these detection challenges, three solutions were implemented, which were color separation, the exclusion of pre-mapping structures, and vertical structure segmentation. For instance, color separation was applied to reduce misclassification between jack supports and visually similar objects such as ceiling pipes, as observed with JS_10. The exclusion of pre-mapping structures mitigated the influence of surrounding object changes, which contributed to the misclassification of JS_14 after the dismantling of JS_16. Finally, vertical structure segmentation improved detection consistency under varying lighting conditions, particularly for JS_20. The effectiveness of these solutions is discussed in greater detail in Chapter 5.

#### Accuracy Evaluation of Jack Support Retention Period Compliance Detection

In this section, the accuracy of the jack support retention period compliance detection process is analyzed based on two criteria. The first criterion measures the daily accuracy from T_1_ to T_28_, and the second evaluates the detection accuracy for each of the 23 jack supports individually. This allows us to assess how consistently the system detected the jack supports and successfully monitored the retention period.

Daily accuracy is calculated based on the proportion of jack supports detected at each time point. For example, at T1, 22 out of 23 jack supports were successfully detected, resulting in an accuracy of 22/23 = 95.65% at that time point. Table 6 below summarizes the daily detection accuracy from T_1_ to T_28_.

Additionally, the accuracy was evaluated based on whether each jack support was successfully detected during its designated retention period. In this case, it was assumed that jack support would no longer be detected after the scheduled dismantling date. The accuracy of the retention period compliance detection for each jack support was calculated as the number of successful detections during the period from T_1_ to T_28_ divided by the total number of days the jack support was supposed to be detected. For example, JS_1 was detected every day from T_1_ to T_28_, so the accuracy is calculated as 28 (detected days)/28 (designated retention period) = 100%. On the other hand, JS_10 was not detected at all due to confusion with the ceiling pipe, so the accuracy is 0% (detected days)/7 (designated retention period) = 0%. Table 7 below summarizes the detection accuracy for each jack support.

Based on a total of 23 jack supports and a 28-day period, the system’s overall accuracy was evaluated. The average detection accuracy over the 28 days is approximately 94.1%, while the average detection accuracy for the 23 jack supports is around 93.4%.

## 5. Discussion

### 5.1. Contribution and Limitations

This study integrates construction drawings, on-site data, and work order documents into a unified jack support monitoring process. By combining these three data sources, the system was able to automate the inspection process through jack support detection and reflection of the work process. This integration contributes to construction management by embedding the monitoring process directly into the workflow, reducing reliance on manual inspections, and minimizing human error.

However, there are several areas that require improvement to enhance system performance. Specifically, detection errors occurred during the detection process of jack supports. For example, as mentioned in the results section, three detection errors occurred during the retention period compliance check. JS_10 was consistently incorrectly detected as a ceiling pipe from T_1_ through T_28_, resulting in a complete failure to detect it. JS_14 was correctly detected until T_14_, but after the dismantling of JS_16, the system began to misidentify a pillar corner as JS_14, causing missed detections from T_15_ through T_28_. Lastly, JS_20 experienced inconsistent detection due to fluctuating lighting conditions at the site, resulting in intermittent detection until its dismantling at T_14_. These issues highlight the need for improvements in the system’s ability to handle environmental changes and object misdetection.

While the model’s 98% accuracy in detecting jack support symbols on construction drawings demonstrates a high level of performance, the 2% error rate remains within a manageable threshold for practical application. In real-world construction scenarios, jack supports are typically installed with a margin of safety, including redundancy to account for unexpected factors or errors. Furthermore, jack supports are designed to distribute loads effectively, minimizing the impact of isolated misclassifications. Therefore, despite the small percentage of errors, the automated monitoring system significantly enhances operational efficiency and safety by reducing the reliance on manual inspections and providing consistent detection accuracy. These improvements outweigh the limitations of the residual error rate, supporting the practical viability of the system’s deployment in construction workflows.

Additionally, the system’s testing was limited to a single project conducted under constrained conditions, which restricts the generalizability of the findings. Diverse site conditions and workflows, which could introduce variability in performance, were not evaluated in this study. Moreover, the study was limited to a 28-day testing period, leaving the long-term reliability and durability of the system unevaluated. Future studies should address these limitations by conducting tests across varied environments and extended timeframes to ensure the system’s adaptability and robustness under practical scenarios.

Another potential improvement involves the integration of RFID (radio frequency identification) technology to enhance the reliability of the matching process between jack supports detected in construction drawings and those identified on-site. By attaching RFID tags to each jack support, unique identifiers could be assigned, enabling precise tracking and matching throughout the construction process. This approach would address challenges such as misclassification or ambiguities in configurations where multiple jack supports share similar spatial layouts. Furthermore, RFID tags could be linked to construction schedules and site plans, providing an additional layer of verification during the detection and monitoring process. Integrating RFID technology has the potential to simplify the matching process, reduce manual verification efforts, and improve the overall accuracy and efficiency of the system in dynamic construction environments. Future research should explore the feasibility of RFID implementation and its integration with the existing object detection framework.

One possible improvement involves the application of advanced detection models, such as YOLOv8. While YOLOv5 demonstrated effectiveness in detecting jack supports, the detection errors observed under challenging site conditions indicate a need for a more robust model. YOLOv8, as an enhanced version of YOLOv5, incorporates advancements in detection accuracy and adaptability to dynamic environments [87,88]. These improvements could address the limitations identified in this study, including misclassification and sensitivity to lighting variations. Future research should conduct a comparative analysis of YOLOv5 and YOLOv8 on similar datasets to assess their respective strengths and determine the most suitable model for construction site monitoring. Such an analysis could refine the detection system and mitigate errors in practical scenarios.

Another potential improvement involves the integration of augmented reality (AR) elements into the proposed system. By incorporating AR, the system could provide real-time visualization of jack support placements and structural compliance at the construction site. This approach would allow users to directly overlay detection results onto the physical environment, enabling immediate identification of discrepancies or errors in jack support installation. Integrating AR could enhance the system’s usability and improve operational efficiency by providing intuitive visual feedback [89,90]. Future research should explore the feasibility of implementing AR technology within the monitoring workflow to further optimize construction site management.

### 5.2. Solution for Preventing the Misdetection of Ceiling Pipes as Jack Supports

In this study, one of the most significant issues that arose during the jack support detection process was the misdetection of ceiling pipes as jack support. In particular, JS_10 was consistently incorrectly detected as a ceiling pipe from T_1_ to T_28_, resulting in a complete failure to detect it throughout the retention period. This misdetection was the most prominent detection error observed in the study, and as such, the focus was placed on resolving this issue. This section presents three key approaches aimed at addressing misdetection, which were then implemented and analyzed in comparison. Each solution was designed to enhance detection accuracy and reduce misdetection through techniques such as color differentiation, the exclusion of pre-mapped structures based on construction plans, and vertical segmentation. By evaluating the effectiveness of each solution, we aim to propose the most efficient strategy for resolving the structural similarities between ceiling pipes and jack supports, thus improving the system’s reliability.

#### 5.2.1. Using a Consistent Color Pattern for Ceiling Pipes

To address the issue of the misdetection of ceiling pipes as jack supports, a color differentiation method was applied. During the detection process, the system often confused ceiling pipes with jack supports due to their similar shape and structure. To mitigate this, red markings spaced at regular intervals were utilized on the pipes. As shown in Figure 12b,c, these red markings helped distinguish between ceiling pipes and jack supports visually. After implementing this method, the detection system’s performance improved and the misdetection rate for JS_10 decreased. As shown in Figure 12a, prior to applying the color pattern, the system mistakenly detected ceiling pipes as jack supports. However, after introducing color differentiation, these issues were reduced. This method, which adds color as an additional detection factor beyond just shape, significantly enhanced the system’s accuracy.

#### 5.2.2. Applying Construction Drawings for Ceiling Pipes

To address the incorrect detection of ceiling pipes as jack supports, construction drawing data were integrated into the detection algorithm to exclude ceiling pipes from the detection targets. Initially, the system struggled due to the structural similarity between ceiling pipes and jack supports, which caused difficulty in distinguishing between them. As shown in Figure 13a, the detection algorithm mistakenly detected ceiling pipes as jack supports, which led to this issue. To address this, construction drawings specifying the positions of ceiling pipes were overlaid onto the detection algorithm. This enabled the detection system to clearly detect ceiling pipes, excluding them from the detection process and allowing it to focus on detecting jack supports. Figure 13b shows how the construction drawings of the ceiling pipes were integrated into the system, reducing the false detections during the monitoring process. Figure 13c shows the system excluding the ceiling pipes from detection. By applying this method, the system was able to filter out unnecessary structures, significantly improving the detection accuracy of jack supports and reducing the likelihood of errors during monitoring.

#### 5.2.3. Jack Support Detection Through Vertical Segmentation

One of the methods implemented to reduce false detections in jack support detection was to segment vertical structures before the detection process. Jack supports are typically installed vertically on construction sites, while other structures, such as ceiling pipes or beams, are often installed horizontally. By prioritizing the separation of vertical structures, the detection algorithm can focus solely on the vertical elements, thereby reducing the risk of mistakenly detecting horizontal structures as jack supports.

The first step in this method is to differentiate vertical and horizontal elements in the images captured on-site. For example, Figure 14a shows the initial image containing both vertical and horizontal structures. The system segments the image into vertical and horizontal elements, leaving only the vertical structures (Figure 14b). During this process, the horizontal structures are excluded from the detection scope, allowing the system to focus exclusively on the vertical structures. In the final step (Figure 14c), the detection algorithm targets only these segmented vertical objects, leading to significantly improved detection accuracy and nearly eliminating false detections caused by horizontal structures.

This approach has been validated in similar studies, where vertical segmentation was proven to be highly effective in improving detection accuracy in construction sites with complex and mixed structural elements [90]. Therefore, focusing solely on vertical elements enables accurate jack support detection and greatly enhances monitoring performance on-site.

#### 5.2.4. Comparison of Three Methods to Prevent Misdetection of Ceiling Pipes as Jack Supports

This section compares and analyzes three main approaches to reduce the misdetection of ceiling pipes as jack supports. The methods include color differentiation, the exclusion of pre-mapped structures, and the segmentation of vertical structures. The evaluation criteria for these methods are based on the false detection reduction rate, ease of algorithm implementation, and processing speed.

First, color differentiation was implemented to distinguish the ceiling pipes from the jack supports by recognizing their color pattern. This method was relatively simple to implement, using 20 images of ceiling pipes and taking about 2.5 h. Due to its simplicity, it was ranked first in algorithm implementation convenience, with a processing speed of 15 ms. However, the false detection reduction rate was 68%, indicating that performance might be limited in complex environments. The second method, excluding mapped structures, used drawing data to exclude ceiling pipes from detection targets. This method involved more complex data matching processes. It required 40 images of ceiling pipes and ceiling piping drawing data, taking around 6 h to implement. As a result, it ranked third in implementation convenience, with the slowest processing speed of 20 ms, but it still achieved a relatively good false detection reduction rate of 78%. Finally, the segmentation of vertical structures prioritized separating vertical structures for detection based on shape and orientation. This method used 20 images of vertical structures and took approximately 3 h and 15 min to implement. It ranked second in implementation convenience and achieved the fastest processing speed at 12 ms, making it suitable for on-site monitoring. It also achieved the highest false detection reduction rate of 86%. Table 8 summarizes the evaluation of each method.

In summary, after evaluating the three methods, the segmentation of vertical structures demonstrated the highest false detection reduction rate and the fastest processing speed, with relatively good algorithm implementation convenience. Therefore, it was selected as the most suitable method for detecting jack supports in complex construction environments, where there is potential confusion with ceiling pipes and other similar structures.

## 6. Conclusions

This study proposed a framework for automatically monitoring the installation and retention period of jack supports at construction sites. The system integrates object detection to recognize both actual jack supports and their corresponding symbols in construction drawings while combining OCR and NLP technologies to extract critical information related to jack supports. The focus of this system was to enhance efficiency and accuracy in the installation and dismantling process of jack supports. By integrating construction drawings, on-site data, and work instructions, the system was able to track and monitor the installation and dismantling of jack supports, thereby reflecting the workflow. This integrated system significantly improves the reliability of work management on construction sites and enhances site safety. The results of the study indicated that the accuracy of detecting jack support compliance with retention periods averaged 94.1% per day, and the average detection accuracy for each jack support was 93.4%. These findings demonstrate the feasibility of accurate management through monitoring.

Although false detections occurred with JS_10, JS_14, and JS_20 during the study, the use of color differentiation, the exclusion of mapped structures, and the segmentation of vertical structures effectively resolved these issues. These methods significantly reduced the rate of false detections, with the vertical structure segmentation achieving an 86% reduction in false detections and the fastest processing speed, making it the most suitable method for detecting jack supports.

However, this study also has several limitations. One key issue is the potential degradation of detection performance due to environmental changes. Future research should focus on overcoming these limitations by integrating more advanced machine learning models and sensor technologies, allowing the system to adapt to various environmental changes. Moreover, long-term testing across diverse construction environments is necessary to validate the system’s robustness and practical applicability. By addressing these challenges, the proposed system could become a vital tool for enhancing the reliability of detection and monitoring tasks in construction, ultimately improving site management efficiency.

## Figures and Tables

**Figure 1 sensors-25-00574-f001:**
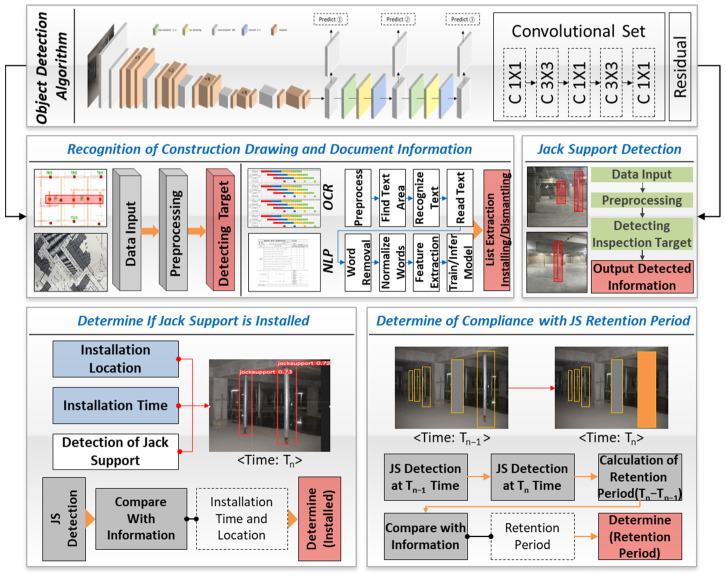
Research framework.

**Figure 2 sensors-25-00574-f002:**
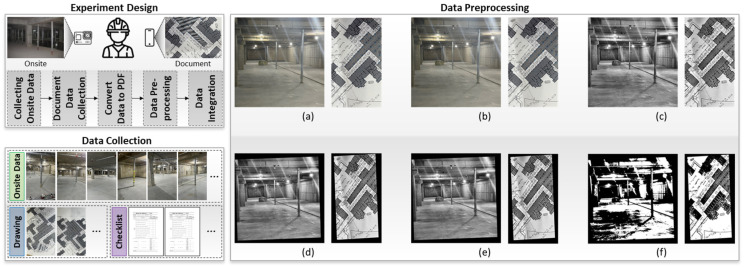
Experiment steps and data preprocessing: (**a**) original image; (**b**) blurred image; (**c**) contrast-enhanced image; (**d**) aligned image; (**e**) normalized image; and (**f**) segmented image.

**Figure 3 sensors-25-00574-f003:**
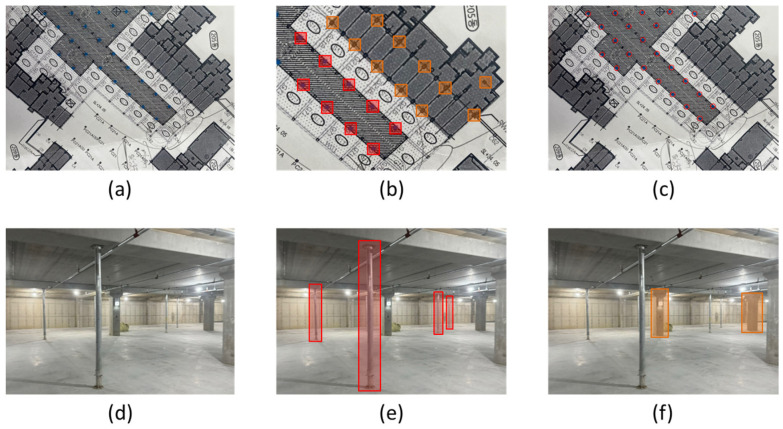
Drawing and on-site detection process: (**a**) jack support on drawing; (**b**) visual similarity between jack support and column symbols in drawing; (**c**) classification of jack support on drawing; (**d**) on-site jack support; (**e**) classification of on-site jack supports; and (**f**) classification of on-site pillars.

**Figure 4 sensors-25-00574-f004:**
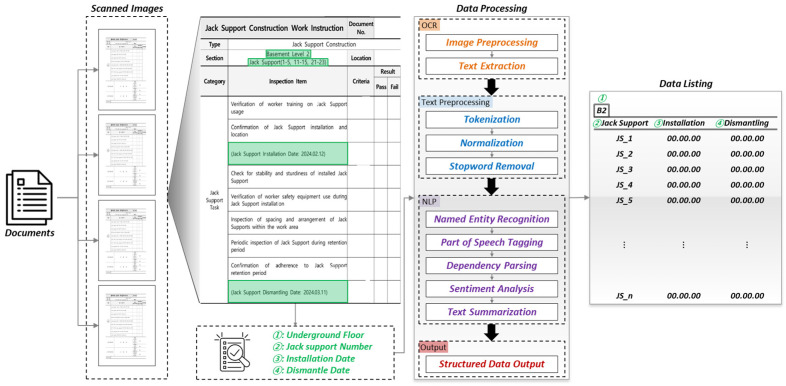
Document information recognition flow.

**Figure 5 sensors-25-00574-f005:**
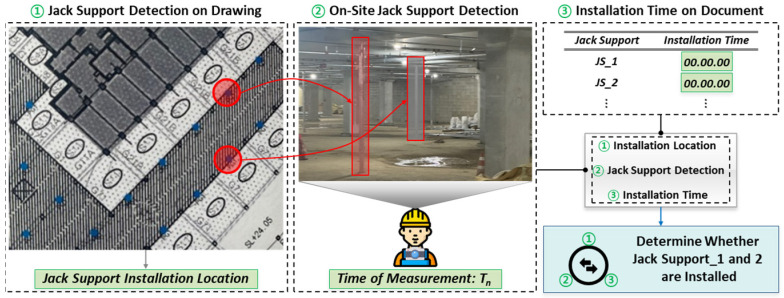
Scenarios to determine whether to install jack support.

**Figure 6 sensors-25-00574-f006:**
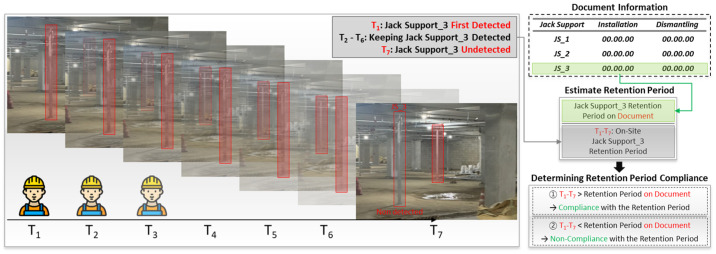
Scenarios for determining jack support retention period compliance.

**Figure 7 sensors-25-00574-f007:**
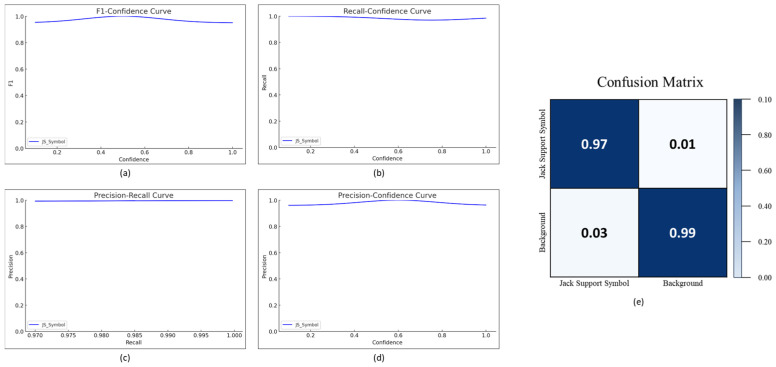
Performance evaluation of drawing object detection model: (**a**) F1 confidence curve; (**b**) recall–confidence curve; (**c**) precision–recall curve; (**d**) precision–confidence curve; and (**e**) jack support detection confusion matrix on drawing.

**Figure 8 sensors-25-00574-f008:**
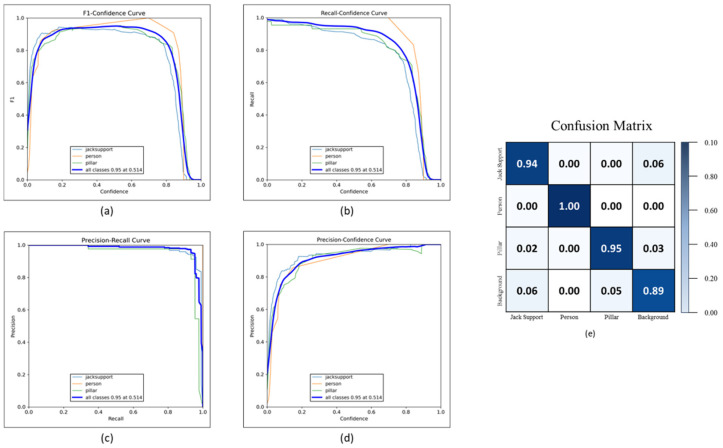
Performance evaluation of on-site object detection model: (**a**) F1 confidence curve; (**b**) recall–confidence curve; (**c**) precision–recall curve; (**d**) precision–confidence curve; and (**e**) on-site jack support detection confusion matrix.

**Figure 9 sensors-25-00574-f009:**
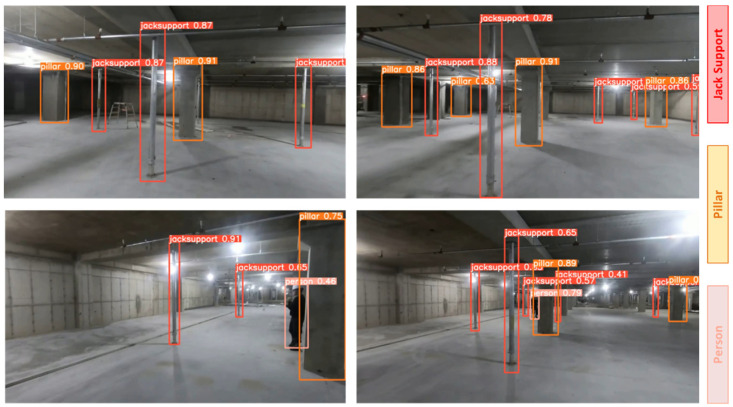
On-site jack support detection result.

**Figure 10 sensors-25-00574-f010:**
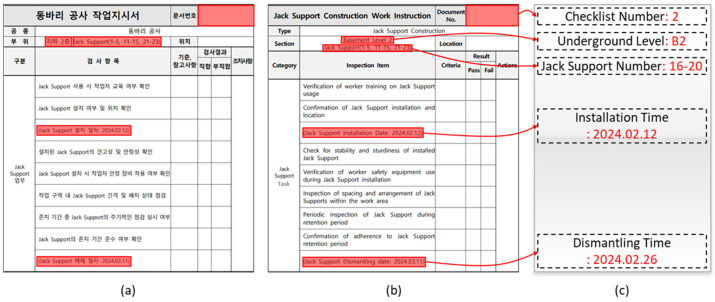
Checklist checking element: (**a**) checklist Korean version; (**b**) checklist English version; and (**c**) extracted information.

**Figure 11 sensors-25-00574-f011:**
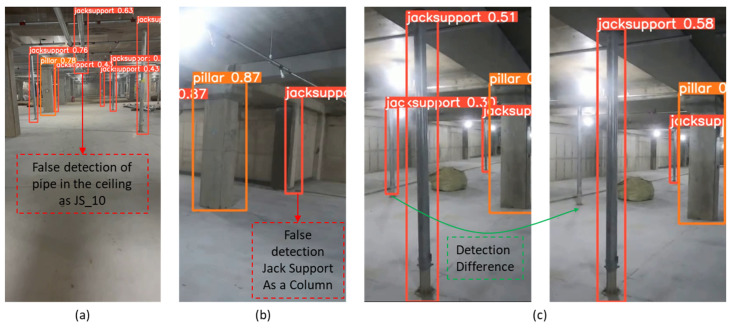
Jack support detection error: (**a**) jack support 10 detection error; (**b**) jack support 14 detection error and (**c**) jack support 20 detection error.

**Figure 12 sensors-25-00574-f012:**
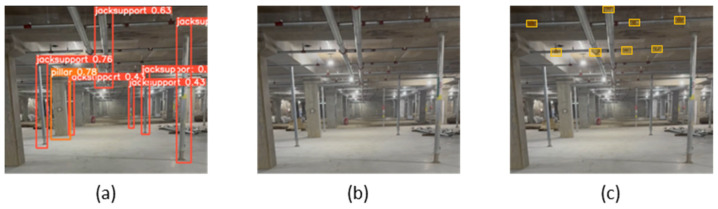
Jack supports and ceiling pipe detection error: (**a**) ceiling pipe detection error; (**b**) on-site image of jack supports and ceiling pipes; and (**c**) red markings on ceiling pipes.

**Figure 13 sensors-25-00574-f013:**
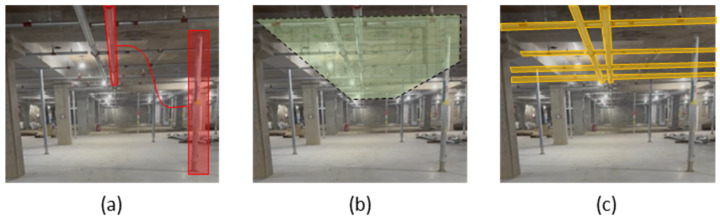
Excluding ceiling pipe detection: (**a**) similarity of jack support and ceiling pipe; (**b**) laying out a ceiling pipe drawing and (**c**) excluding ceiling pipes from detection.

**Figure 14 sensors-25-00574-f014:**
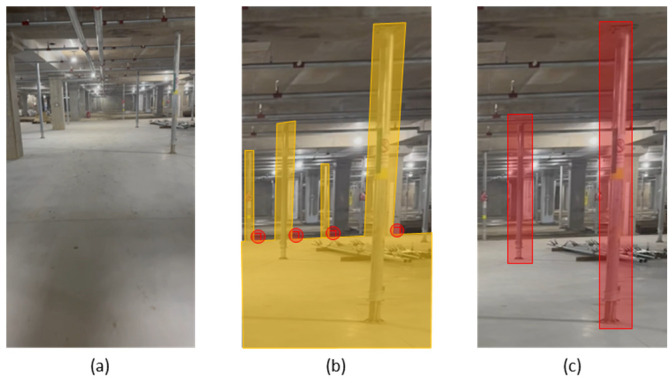
Vertical materials detection process: (**a**) on-site jack supports and pipes in ceiling; (**b**) segment materials perpendicular to the floor; and (**c**) detection jack support within segmented materials.

**Table 1 sensors-25-00574-t001:** Text preprocessing steps.

Text Preprocessing	Purpose	Details
Tokenization	− Breaks text into smaller, manageable units	− Divides text from construction documents (e.g., “jack support”, “installation date”) into smaller units like keywords and dates. This enables easier comparison between on-site data and construction schedules.
Normalization	− Ensures consistency across varying text formats	− Converts all characters to lowercase and removes irrelevant punctuation (e.g., “#”, “!”), ensuring consistency when comparing project plans and real-time on-site data.
Stopword Removal	− Focuses analysis on meaningful keywords	− Eliminates frequent, non-informative words like “and”, “the”, and “is” to highlight critical terms such as “jack support” and “installation date” for accurate data analysis.

**Table 2 sensors-25-00574-t002:** NLP technique details.

NLP Techniques	Purpose	Details
Named Entity Recognition (NER)	− NER identifies key entities within construction-related text, such as dates, locations, and specific identifiers like “jack support numbers”.	− This technique is critical for extracting relevant information from documents, such as installation schedules or maintenance logs, and linking it to on-site data.
Part-of-Speech Tagging	− Labels each word in the text with its part of speech (e.g., noun, verb, adjective).	− Distinguishes between actions (e.g., “install,” “dismantle”) and objects (e.g., “jack support”, “retention period”) in construction work instructions.
Dependency Parsing	− Analyzes sentence structure to identify relationships between words.	− Identifies relationships like “installation or dismantling date,” ensure that the installation and removal of jack supports align with project schedules.
Sentiment Analysis	− Assesses the emotional tone or sentiment in the text.	− Can detect concerns or issues in the inspection results, such as “failure” or “pass” for the stability or arrangement of jack supports.
Text Summarization	− Condenses lengthy text into a concise summary.	− Summarizes work instructions or inspection logs, such as highlighting important dates like “Jack Support Dismantling Date: 11 March 2024”.

**Table 3 sensors-25-00574-t003:** Checklist recognition result.

Checklist Number	Jack Support Number	Installation Time	Dismantling Time
Checklist_1	JS_1 to 12	○	○
Checklist_2	JS 13 to 23	○	○
Checklist_3	JS_24 to 33	○	○
⋮	⋮	⋮	⋮
Checklist_17	JS_169 to 181	○	○

**Table 4 sensors-25-00574-t004:** Evaluation results of jack support installation.

Jack Support Number	Detected in Drawing	Detected Onsite at T_1_	Installation Time	Error	Determine
JS_1	○	○	12 February 2024	-	√
JS_2	○	○	12 February 2024	-	√
JS_3	○	○	12 February 2024	-	√
⋮	⋮	⋮	⋮	⋮	⋮
JS_10	○	×	12 February 2024	○*_a)_	×
⋮	⋮	⋮	⋮	⋮	⋮
JS_23	○	○	12 February 2024	-	√

**Table 5 sensors-25-00574-t005:** Result of determining retention compliance T_1_: 12 February 2024, T_2_: 13 February 2024 ⋯ T_28_: 11 March 2024.

Jack Support	Dismantling Time	T_1_	T_2_	T_3_	T_4_	T_5_	T_6_	T_7_	T_8_	T_9_	T_10_	T_11_	T_12_	T_13_	T_14_	T_15_	T_16_	T_17_	T_18_	T_19_	T_20_	T_21_	T_22_	T_23_	T_24_	T_25_	T_26_	T_27_	T_28_	Error	Determine
JS_1	T_28_	○	○	○	○	○	○	○	○	○	○	○	○	○	○	○	○	○	○	○	○	○	○	○	○	○	○	○	○	-	√
JS_2	T_28_	○	○	○	○	○	○	○	○	○	○	○	○	○	○	○	○	○	○	○	○	○	○	○	○	○	○	○	○	-	√
JS_3	T_28_	○	○	○	○	○	○	○	○	○	○	○	○	○	○	○	○	○	○	○	○	○	○	○	○	○	○	○	○	-	√
JS_4	T_28_	○	○	○	○	○	○	○	○	○	○	○	○	○	○	○	○	○	○	○	○	○	○	○	○	○	○	○	○	-	√
JS_5	T_28_	○	○	○	○	○	○	○	○	○	○	○	○	○	○	○	○	○	○	○	○	○	○	○	○	○	○	○	○	-	√
JS_6	T_7_	○	○	○	○	○	○	○	-	-	-	-	-	-	-	-	-	-	-	-	-	-	-	-	-	-	-	-	-	-	√
JS_7	T_7_	○	○	○	○	○	○	○	-	-	-	-	-	-	-	-	-	-	-	-	-	-	-	-	-	-	-	-	-	-	√
JS_8	T_7_	○	○	○	○	○	○	○	-	-	-	-	-	-	-	-	-	-	-	-	-	-	-	-	-	-	-	-	-	-	√
JS_9	T_7_	○	○	○	○	○	○	○	-	-	-	-	-	-	-	-	-	-	-	-	-	-	-	-	-	-	-	-	-	-	√
JS_10	T_7_	×	×	×	×	×	×	×	-	-	-	-	-	-	-	-	-	-	-	-	-	-	-	-	-	-	-	-	-	○*_a)_	×
JS_11	T_28_	○	○	○	○	○	○	○	○	○	○	○	○	○	○	○	○	○	○	○	○	○	○	○	○	○	○	○	○	-	√
JS_12	T_28_	○	○	○	○	○	○	○	○	○	○	○	○	○	○	○	○	○	○	○	○	○	○	○	○	○	○	○	○	-	√
JS_13	T_28_	○	○	○	○	○	○	○	○	○	○	○	○	○	○	○	○	○	○	○	○	○	○	○	○	○	○	○	○	-	√
JS_14	T_28_	○	○	○	○	○	○	○	○	○	○	○	○	○	○	×	×	×	×	×	×	×	×	×	×	×	×	×	×	○*_b)_	×
JS_15	T_28_	○	○	○	○	○	○	○	○	○	○	○	○	○	○	○	○	○	○	○	○	○	○	○	○	○	○	○	○	-	√
JS_16	T_14_	○	○	○	○	○	○	○	○	○	○	○	○	○	○	-	-	-	-	-	-	-	-	-	-	-	-	-	-	-	√
JS_17	T_14_	○	○	○	○	○	○	○	○	○	○	○	○	○	○	-	-	-	-	-	-	-	-	-	-	-	-	-	-	-	√
JS_18	T_14_	○	○	○	○	○	○	○	○	○	○	○	○	○	○	-	-	-	-	-	-	-	-	-	-	-	-	-	-	-	√
JS_19	T_14_	○	○	○	○	○	○	○	○	○	○	○	○	○	○	-	-	-	-	-	-	-	-	-	-	-	-	-	-	-	√
JS_20	T_14_	○	○	×	○	×	×	○	×	○	○	×	○	○	×	-	-	-	-	-	-	-	-	-	-	-	-	-	-	○*_c)_	×
JS_21	T_28_	○	○	○	○	○	○	○	○	○	○	○	○	○	○	○	○	○	○	○	○	○	○	○	○	○	○	○	○	-	√
JS_22	T_28_	○	○	○	○	○	○	○	○	○	○	○	○	○	○	○	○	○	○	○	○	○	○	○	○	○	○	○	○	-	√
JS_23	T_28_	○	○	○	○	○	○	○	○	○	○	○	○	○	○	○	○	○	○	○	○	○	○	○	○	○	○	○	○	-	√

**Table 6 sensors-25-00574-t006:** Daily accuracy of jack support retention period compliance detection.

Time	Detected Jack Supports	Total Jack Supports to Detection	Accuracy (%)
T_1_	22	23	95.65
⋮	⋮	⋮	⋮
T_14_	17	18	94.44
⋮	⋮	⋮	⋮
T_28_	12	13	92.31

**Table 7 sensors-25-00574-t007:** Detection accuracy for each jack support during the designated retention period.

Jack Support	Detected Days	Designated Retention Period	Accuracy (%)
JS_1	28	28	100
⋮	⋮	⋮	⋮
JS_10	0	7	0
⋮	⋮	⋮	⋮
JS_14	14	28	50
⋮	⋮	⋮	⋮
JS_20	8	14	57.14
⋮	⋮	⋮	⋮
JS_23	28	28	100

**Table 8 sensors-25-00574-t008:** Summary of false detection reduction method evaluation.

Method	False Detection Reduction Rate (%)	Algorithm Implementation Convenience (Rank)	Processing Speed
Color Differentiation	68	1	2 (15 ms)
Exclusion of Pre-Mapped Structures	78	3	3 (20 ms)
Vertical Structure Segmentation	86	2	1 (12 ms)

## Data Availability

All data, models, or code generated or used during the study are available from the corresponding author upon request.

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
