# Peer review of "Time-Series Image-Based Automated Monitoring Framework for Visible Facilities: Focusing on Installation and Retention Period"

_sensors, 2025, doi:10.3390/s25020574_

Round 1
Reviewer 1 Report
Comments and Suggestions for Authors
In the peer-reviewed paper “Time-Series Image-Based Automated Monitoring Framework for Visible Facilities: Focusing on Installation and Retention Period”, an integrated monitoring framework combining computer vision-based object detection and document recognition techniques is proposed. The system uses YOLOv5 to detect jack pads on both construction drawings and on-site images captured by wearable cameras. Optical character recognition (OCR) and natural language processing (NLP) extract the installation and dismantling dates of the pads from the working documents. The proposed framework enables continuous monitoring and ensures the adherence to the jack installation dates by matching the on-site data with the documented requirements. To further improve the reliability of the detection, techniques such as color differentiation, construction plan overlays, and vertical segmentation were implemented, which significantly improved the performance of the recognition process. What distinguishes this study is its dual application of object detection - not only on on-site images but also on construction drawings, which ensures the correspondence between the planned and implemented jack installation activities.
The introduction provides a detailed review of literary sources on the research topic, and indicates the proposed methods and approaches of other authors.
The technological aspects of the problem statement and the general principles of applying image recognition mechanisms are described in sufficient detail.
The conclusions obtained in the work are sufficiently substantiated.
The results of the study will be useful to specialists in the field of automation of construction processes using artificial intelligence technologies.
At the same time, there are some comments and suggestions for improvement regarding the article:
1. The article should be supplemented with a concentrated list of practical parameters and settings of the YOLOv5 network for the task of recognizing objects: jacks, columns, pipes, etc.
2. It is necessary to describe in more detail how the jack supports were identified on site and how they were compared with the images of the supports on the construction drawings. At least in this process there should be a reference to the floor, and if there are rooms with similar configurations inside the floor, to some marks on the floor. Perhaps in this case, the use of RFID tags will be useful
3. In Fig. 4, 2 languages ​​are used for the text, which worsens the perception of information.
4. It would be of interest to conduct a comparative analysis of the application of YOLOv5 and YOLOv8 networks for the specified tasks.
5. A natural extension of the proposed system is the implementation of augmented reality elements in it. Perhaps this should be indicated in the article with a list of additional features.
If these comments are corrected, the article may be accepted for publication in the journal Sensors.

Author Response
Dear Reviewer 1,
We sincerely appreciate your thorough and insightful review of our manuscript. Your comments and suggestions were invaluable in improving the clarity, technical depth, and overall quality of our work.
We have carefully addressed each of your comments, and a detailed response has been provided for every point in the attached file. We hope these revisions align with your expectations and clarify any ambiguities that were present in the initial submission.
Thank you once again for your valuable feedback and for contributing to the enhancement of our research. Please feel free to reach out if there are any additional concerns or suggestions.

Reviewer 2 Report
Comments and Suggestions for Authors
This paper addresses a critical issue in automating the installation and retention monitoring of Jack Supports at construction sites and records notable technical achievements. However, the study exhibits some limitations, such as a lack of justification for the Confidence Threshold settings, insufficient testing across diverse environments, and a lack of contextual relevance for the accuracy metric in practical domains. Resolving these issues could elevate the study's practical and academic contributions.
1. Insufficient Technical Detail on Jack Support ID Matching in ‘3. Methodology’
The technical explanation of the process for matching Jack Supports detected in construction drawings with those detected on-site is inadequate. There is a lack of detailed descriptions regarding the algorithm, data matching logic, and error handling methods employed, which raises concerns about the reproducibility of the results. It is unclear how the identity of Jack Supports is verified, whether through location, timestamps, or additional unique identifiers. For example, more specific information about the criteria used (e.g., spatial coordinates or timestamps) to match Jack Supports between drawings and on-site detections would significantly enhance the credibility of the paper.
The same issue applies to the ID matching process during data collection and preprocessing. The criteria for matching construction drawings and on-site images (e.g., shared spatial references) are not explicitly detailed.
2. Insufficient Connection Between ‘4. Result’ and ‘5. Discussion’
The three solutions presented in Chapter 5, ‘colour separation,’ ‘exclude pre-mapping structure,’ and ‘vertical structure segmentation,’ appear suddenly. Result without sufficient explanation of the related experimental results. Therefore, it is recommended that the experiment results of each solution for the specific cases (JS_10, JS_14, and JS_20) that were problematic in ‘4. Result’ are mentioned and then further discussed in ‘5. Discussion’.
3. Limited Testing in Diverse Environments
While this limitation is briefly mentioned in ‘6. Conclusion,’ it should be addressed more comprehensively in ‘5. Discussion’ to ensure the objectivity of the findings. The paper should acknowledge that the study was conducted in a single project under constrained conditions and that tests in various environments (e.g., different site conditions or workflows) were not performed. This limitation restricts the generalizability of the results.
Additionally, the study was limited to a 28-day testing period, leaving the long-term reliability and durability of the system unevaluated.
4. Lack of Justification for Optimal Confidence Threshold Settings
Although the F1-Confidence Curve and other metrics are mentioned as performance indicators, the Confidence Threshold settings used to achieve the reported 98% accuracy are not explicitly justified. In Line 360, it is noted that “These curves help assess the model’s accuracy and robustness across different confidence thresholds,” but there is no further discussion about how the Confidence Threshold was optimized. This omission undermines the reliability of the model’s performance evaluation. Since the Threshold significantly impacts model performance, a clear explanation of how it was experimentally optimized across various datasets is necessary.
5. ‘5. Discussion’ requires contextual analysis to explain the practical meaning of accuracy indicators in actual construction site management
While the 98% accuracy achieved by the model is commendable, the paper lacks a contextual analysis of what this accuracy means in terms of construction site safety and operational efficiency. For example, the paper could explore how a 98% accuracy rate reduces installation errors for Jack Supports. In a project where 100 Jack Supports are detected daily, a 98% accuracy would result in an average of 2 errors per day. Discussing how this aligns with safety management goals would enhance the paper’s academic and practical value.
6. The titles 4.1. Object Detection Results and 4.2. Object Detection Results are identical. This duplication may confuse readers and fails to clearly differentiate the focus of each section.
Author Response
Dear Reviewer 2,
We sincerely appreciate your thorough and insightful review of our manuscript. Your comments and suggestions were invaluable in improving the clarity, technical depth, and overall quality of our work.
We have carefully addressed each of your comments, and a detailed response has been provided for every point in the attached file. We hope these revisions align with your expectations and clarify any ambiguities that were present in the initial submission.
Thank you once again for your valuable feedback and for contributing to the enhancement of our research. Please feel free to reach out if there are any additional concerns or suggestions.

Round 2
Reviewer 1 Report
Comments and Suggestions for Authors
I believe that in this form the article can be published in the journal Sensors.
Reviewer 2 Report
Comments and Suggestions for Authors
The revised manuscript demonstrates significant improvements, particularly in clarifying the section titles, which now effectively differentiate between object detection results from construction drawings and on-site images. This change enhances the readability and structure of the paper, addressing previous concerns regarding potential reader confusion.
Overall, the study provides a valuable contribution to automated monitoring in construction, and the integration of object detection with OCR and NLP is well-articulated. The detailed discussion on addressing detection errors, along with the comparative analysis of proposed solutions, adds depth to the work.